# Using Sustainability-Oriented Developer Obligations and Public Land Development to Create Public Value

Melissa Candel 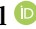

Department of Real Estate and Construction Management, Division of Real Estate Planning and Land Law, KTH Royal Institute of Technology, 11428 Stockholm, Sweden; candel@kth.se; Tel.: +46-7684-57979

**Abstract:** Swedish municipalities use negotiable developer obligations and public land development in sustainability-profiled districts to achieve various public sustainability objectives. They initiate and govern these districts, which act as models for sustainable urban development and testbeds for new sustainability-related policies, using municipally owned land. Public land development in Sweden enables municipalities to include sustainability-oriented negotiable developer obligations in development agreements. The aim of the study is to investigate how Swedish municipalities use sustainability-oriented negotiable developer obligations together with public land development, and to identify what public value outcomes they currently seek to create by using these public value capture instruments. Sustainability-oriented negotiable developer obligations are investigated in relation to municipalities' desired public value outcomes in five sustainability-profiled district developments in different Swedish municipalities. Findings illustrate that Swedish municipalities use negotiable developer obligations to create ecological, social and cultural, political, and economic public value outcomes. This calls for more research investigating different forms of value and value creation in relation to public value capture instruments.

**Keywords:** negotiable developer obligations; public value capture; public value; public land development; municipal land allocations; sustainable urban development

## 1. Introduction

Public land development is often seen as a source of revenue for municipalities [1]. It is defined herein as a process in which public authorities produce building plots, using land they acquire and own, and transfer them to building developers [2]. Public value capture in public land development is typically achieved by selling the building plots, and a common objective is to finance public urban infrastructure (e.g., [3–5]). According to Munoz Gielen and van der Krabben [6], public urban infrastructure includes climate adaptation and mitigation, affordable and social housing, as well as roads, public spaces and public facilities. Another public value capture instrument used to finance such public urban infrastructure is negotiable developer obligations (NDOs) [7], which is receiving more attention as relying on private financing is becoming more common [6]. In Sweden, municipalities sometimes combine the use of these public value capture instruments to drive sustainable development, a practice investigated in this paper.

Valtonen et al. [5] examine how different public objectives are achieved in public land development in Finland and Sweden. The public objectives they focus on are economic sustainability in the form of public cost recovery, environmental sustainability, and social sustainability in the form of treating landowners equitably. However, current practices observed in Swedish sustainability-profiled district developments (see [8]) and developments on municipal land in general (see [9]) suggest that NDOs are used in combination with public land development to achieve a much wider variety of sustainability-related public objectives. In Sweden, it is specifically the utilization of public land that enables municipalities to prescribe sustainability-oriented NDOs for building development projects. They are used to encourage property developers to create specific contributions that add

value to the public sphere, which include, but are not limited to, public urban infrastructure provision.

In the growing stream of literature on public value within the field of public management, contributions that add value to the public sphere are considered public value outcomes [10–12]. According to Benington [13,14], public value outcomes encompass ecological, social and cultural, political and economic dimensions. From the previous literature on developer obligations, it is not clear how this public value capture instrument is used to create different dimensions of public value. The aim of the study is to investigate how Swedish municipalities use sustainability-oriented NDOs to create public value, and to identify what specific public value outcomes they are pursuing. This is done by the following two research questions: (1) How are sustainability-oriented NDOs used in Swedish sustainability-profiled districts to create public value?; (2) What types of public value outcomes are Swedish municipalities currently pursuing through the use of sustainability-oriented NDOs?

Findings are based on a multiple case study of five sustainability-profiled district developments in both larger and smaller municipalities throughout Sweden. The focus in the study is on these types of districts since they have especially high ambitions in terms of sustainable development and innovation. Although they are not indicative of most urban development projects in Sweden, it has become an increasingly common practice for municipalities to use public land to initiate and govern sustainability-profiled district developments that are to act as testbeds for innovation and models for sustainable urban development (see, e.g., [15]). The investigation also specifically focuses on municipalities' objectives when utilizing these public value capture instruments and not on the actual outcomes, which should be further explored in future research.

Previous literature on developer obligations as public value capture instruments has typically focused on the provision of infrastructure, such as roads, surrounding private properties (e.g., [6,16–18]). The present study contributes to these ongoing discussions by exploring sustainability-oriented NDOs that instead entail the implementation of sustainable and innovative solutions and practices within private property, such as wastewater recycling systems. Although local authorities prescribe them to create public value beyond those individual properties, it is questionable whether they should be considered contributions to public urban infrastructure. Contributions are also made to the literature on public land development. The majority of this previous literature comes from the Netherlands and is focused on their institutional framework (e.g., [4,19,20]). However, public land development is also used in varying degrees in the Nordic countries including Sweden, Finland, Denmark and Norway, (see, e.g., [2,5,21–23]), as well as in Switzerland [24], Austria [25] and the United States [26,27]. Then, there are also several countries where most of the land is publicly owned, such as Singapore [28,29], China [30,31] and Hong Kong [32–34]. This paper contributes to public land development literature by presenting empirical cases from the Swedish context, as well as investigating trends in current practices and discussing how they might offer valuable insights for other countries.

The paper is structured as follows. The next section elaborates on the theoretical concepts of public value capture, negotiable developer obligations and public value outcomes. This is followed by a description of the Swedish context. The case studies that were carried out and the methods for gathering and analyzing materials are then described, followed by a presentation of the results. The subsequent discussion investigates the theoretical implications of expanding the conception of value in relation to public value capture using public value creation theory. The paper concludes with the main theoretical contributions, implications for policy and for practitioners, limitations and suggestions for future research.

## 2. Theoretical Framework

### 2.1. Public Value Capture Instruments

The term public value capture is most commonly used to denote methods that public authorities use to capture unearned land value increments from landowners [5,35–37],

although unearned value increases in real estate are often considered as well [6,18]. Unearned land value increments refers to increases in land value not caused by landowners but by some form of public action, such as public planning decisions, changes in land-use regulations, public investments in infrastructure and public services, population growth, or economic development [5,38,39]. However, in practice, it is not easy to distinguish between earned and unearned land value increments, mainly because different phases in property development all entail varying increases in property value caused by different factors and actors [37,40]. As a result, there is typically an uncertainty over what actor/actors have caused what value increase and thereby who should capture what value from property developments [6,16,36,38]. Christensen [41] illustrate the difficulty in accurately identifying value increments for different planning and development stages, which Valtonen et al. [5] highlight as an issue for equitable public value capture. This can be an especially problematic issue for justifying the use of direct public value capture instruments.

Direct instruments are public value capture instruments that are solely based on the motivating rationale that unearned land value increments should be redistributed to the community [6,18,37], or in other words, to the public. This is typically accomplished by some form of taxation. Indirect instruments, on the other hand, do not hinge on linking specific value increments to public action in the same way as they build on a variety of other motivating rationales [6,37]. While there might be an element of capturing unearned increments, this can be much less explicit in the case of indirect instruments and may even be concealed by other motivating rationales brought to the fore as the primary objectives [5,18,37]. Examples of other motivating rationales include cost recovery, a need for resources to provide public services and the internalization of costs to mitigate impacts and negative externalities [6,18,37]. The use of indirect instruments in some empirical contexts might be difficult to connect directly to capturing unearned private land value gains, and may instead be capturing value increments caused by the landowners. For this reason, some have suggested that public value capture should more broadly refer to any instrument that captures any increase in land and building value, and not solely unearned land value increments [6,7], a perspective adopted here as well. As a result of being more pragmatic, flexible and adaptable in practice, indirect instruments are more common than direct instruments. However, both direct and indirect public value capture instruments are sometimes prescribed within the same projects [37].

While public value capture is typically concerned with the impact of public action on private land values, there is also the case of public land development to consider here. In addition to direct and indirect instruments, Alterman [37] categorizes active land policy regimes, such as public land banking and development (see Figure 1 for more examples), as macro value capture instruments. An active approach to land development entails local authorities purchasing and assembling land, developing it, providing infrastructure and selling the serviced building plots to property developers for building development [2, 18,37]. Value increases from public investments in infrastructure and development rights are captured by being reflected in the land sale price [18,37] or through the prescription of contributions that are leveraged for the sale [6]. Public land sales are recognized as a source of public revenue that governments can use to finance public urban infrastructure and public services in several countries [1,26], especially in countries where much of the land is publicly owned [33]. Owning land puts public bodies in stronger positions for negotiations with developers. Public land development does however also mean that public bodies assume financial risks and the ability to finance public urban infrastructure using this form of public value capture is highly dependent on the housing market [18].

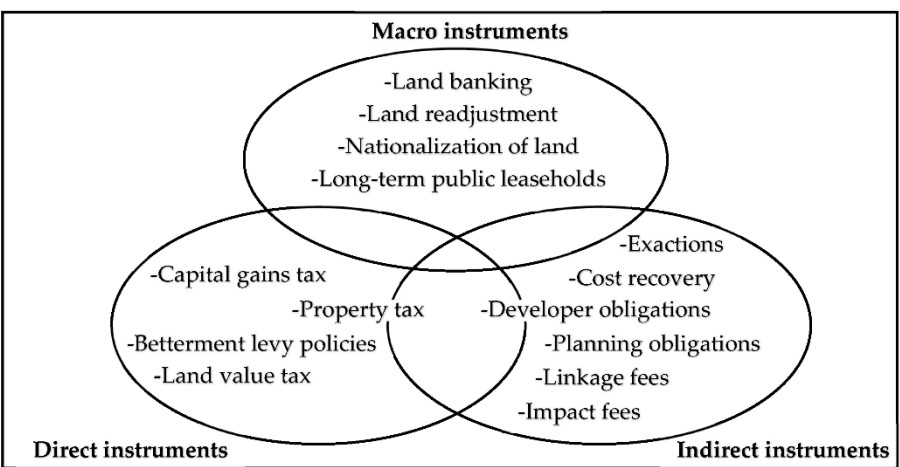

**Figure 1.** Examples of direct, indirect and macro public value capture instruments (source: the author, based on examples from [6,37]).

### 2.2. Negotiable Developer Obligations

Developer obligations are typically considered an indirect public value capture instrument used to finance public urban infrastructure by placing requirements on developers [6,17,18]. However, depending on the motivating rationale that is used, they can also be applied as direct instruments [18]. This instrument has a variation of names used in different countries, such as 'planning gain' in Great Britain [42,43] or 'exactions' in the United States [44], but 'developer obligations' is recognized as a general term used internationally [17,37,44] and will therefore be used here as well. Gozalvo Zamorano and Muñoz Gielen [16] (p. 278) define developer obligations as "contributions of property developers and landowners made in exchange for a public decision on land-use regulations that increases the economic value of their properties". Land-use regulations can here include "additional development rights, fast-track processing, or relaxation of some regulation" [37] (p. 775), and 'contributions' typically refer to some form of monetary payment, land, or construction services for the direct provision of infrastructure [17,18]. However, requirements that entail other types of contributions from property developers could potentially be included within the scope of this definition. One example is the implementation of sustainable and innovative solutions within private properties that are intended to create public value beyond those individual properties, which is the focus of the investigation herein.

Crow [42] (p. 361) suggests that an examination of developer obligations should include an investigation of both the *product*, referring to the "nature and purpose of the gain sought or offered", and the *process* of using them in practice. In relation to the process, a distinction is typically made between non-negotiable developer obligations (N-NDO) and negotiable developer obligations (NDO) [6,17], the latter being the focus in the study presented here. As implied by their names, N-NDOs are not negotiable while NDOs are negotiated between local planning authorities and developers. N-NDOs are prescribed in national and regional legislation and in more detail in local policy documents and legally binding land use or zoning plans [16,18] and have generally been explored more than NDOs in previous literature [17]. NDOs are often less regulated and thereby provide planning flexibility, which is important for dealing with high levels of complexity [17]. In their study from the Netherlands, Munoz Gielen and Lenflerink [18] found that NDOs are easy for practitioners to prescribe since they typically do not require detailed legislative support. Furthermore, Turk [17] found that NDOs in Turkey have a higher public value capture capacity than N-NDOs. However, he also found that they are coupled with lower levels of transparency and accountability resulting in increased uncertainty and risk for developers, which has been observed in other national contexts as well [7,18].

Several scholars have previously questioned and argued against the use of NDOs (see, e.g., [21,42,43]), predicated on the principle that planning permissions made by local planning authorities should not be bought and sold. This raises the question of legitimate and good practice. In light of the numerous political and legal issues and concerns over misuse, bias and unequal treatment typically voiced in discussions about NDOs (e.g., [17, 18,37]), Hendricks et al. [7] argue that it is important to have clear and reasonable criteria for the limits of their use. In the case of public land development, Walters [1] (p. 8) argues that developer obligations can be seen as a way of converting public "land assets to infrastructure assets" meaning they are used more as cost recovery mechanisms and less for capturing unearned private land value gains. As discussed previously, public land development is a macro public value capture instrument [37], and NDOs are a form of indirect instruments, meaning the use of NDOs together with public land development technically entails utilizing two different types of value capture instruments in the same project. A pertinent question is thereby whether this is resulting in a double levy.

Capturing increases in land and building value, in the form of public revenue, is a current and pressing issue with European municipalities' decreasing economic and financial means to provide public urban infrastructure and public services [7]. However, value entails more than revenue. For example, Heeres et al. [45] considered other forms of value that are created as a result of value capturing. They argue that value capturing has both financial value and cooperative value in the form of enhanced cooperation between fragmented actors. While value capturing might create other forms of value, it is uncertain whether the concept of public value capture can be extended to also include capturing other forms of value. Other potential types of value created by the use of different public value capture instruments do, however, warrant further investigation.

*2.3. Public Value*

The growing stream of literature on public value creation in the field of public management offers a broader conception of public value than simply public revenue. This conception of public value is used here as a theoretical framework to analyze NDOs that entail different types of contributions to the public sphere. The rise of the concept is largely accredited to Mark Moore and his seminal book *Creating Public Value: Strategic Management in Government* [46] in which he proposes that the primary role of public organizations should be to create public value. Over the years, different meanings and conceptualizations of public value have evolved, making it important for researchers to be explicit about what they mean when they use the concept [10,47]. Firstly, a distinction is typically made between public value, which is created, and public values, which are achieved [48].

According to Bozeman [49] (p. 13), public values refers to the rights and responsibilities of citizens in a society and "the principles on which governments and policies should be based", over which there is a normative consensus. Such public values, seen at the societal level, should be identifiable in, for example, policies, constitutions, and legislative mandates [50]. In their study, Jørgensen and Bozeman [50] categorize the most central of these public values in the US, UK and Scandinavia into human dignity, sustainability, citizen involvement, openness, secrecy, compromise, integrity, and robustness. When considering different types of public values, a distinction is often made between procedural and performance public values, which are normative and process-related, and substantive public values, which are more sector-specific objectives to provide certain products and services [51,52]. Procedural values, such as lawfulness and transparency, are related to the quality of governance processes, while performance values are related to effectiveness and efficiency in delivering public services.

Public values are closely related to the public interest, which refers to outcomes for a society's well-being and long-term survival [49]. However, public values are measurable and something that the public sector can achieve [46,53], while the public interest is an ideal to be pursued [49]. However, determining how public values should be measured is disputed [48], and actually measuring them can be difficult in practice [47]. For example,

while Moore [46] and Bozeman [49] consider public values as objective, Meynhardt [53] believes that public value resides in the subjectively assessed quality of the relationship between individuals and a society and argues that their creation, or diminishment, should thereby be assessed intersubjectively.

While the literature on public values focuses on the societal level, the public value creation literature is more concerned with the actors that are creating public value, such as public managers and public organizations [46] or partnerships and networks [54]. In this literature, public value is typically defined as contributions to the public sphere that are valued by the citizenry [12–14], which is the definition adopted in this study. Public value is, in other words, both "what the public values" and "what adds value to the public sphere" [14] (p. 42). What is valued can be seen as inputs from the public, which consists of many diverse groups, while public benefits are outputs or outcomes for society [14]. These two dimensions are not always perfectly aligned with each other and can thereby be studied separately to some degree. In the study presented herein, the focus is mainly on the outcomes for society that municipalities have determined will add value to the public sphere, and not on the democratic processes of determining what the public actually values. Benington [13,14] breaks down public value outcomes into four dimensions of value that he suggests add value to the public sphere, summarized and defined in Table 1.

**Table 1.** Dimensions of public value and their definitions.

| Dimensions of Public Value | Definitions |
| --- | --- |
| Ecological value | "adding value to the public realm by actively promoting sustainable development and reducing public 'bads' like pollution, waste, global warming" [1] |
| Social and cultural value | "adding value to the public realm by contributing to social capital, social cohesion, social relationship, social meaning and cultural identify, individual and community well-being" [1] |
| Political value | "adding value to the public realm by stimulating and supporting democratic dialogue and active public participation and citizen engagement" [1] |
| Economic value | "adding value to the public realm through the generation of economic activity, enterprise and employment" [1] |

[1] [14] (pp. 45–46).

According to Benington [14], economic public value does not refer to public revenue but rather entails generating economic activity in the public sphere. To prevent confusion, distinctions will be made between public revenue and economic public value throughout the rest of the paper. The ecological, social and cultural, and the political public value dimensions are, however, closely related to environmental and social sustainability in the built environment. Environmental sustainability and ecological public value are both related to reducing negative impacts on the environment. Social sustainability is more difficult to define, but is typically associated with a variety themes such as equitable access, democracy and participation, safety, and social capital within the community [55–57]. According to Bovaird and Loeffler [58], creating political public value includes fostering trust, legitimacy and efficiency in decision-making processes. Social sustainability in the built environment thereby encompasses both the social and cultural and the political public value dimensions. Public value can also be created as a result of enhancing existing public value creation processes through innovation and improvement. Hartley [59] argues that innovations, as opposed to continuous and incremental improvements, can lead to significant improvements of public services that contribute to public value creation. However, she also points out that innovation in and of itself does not always result in improvements and can in some cases even detract from public value creation. Unsuccessful innovation attempts can however still be valuable learning opportunities [59].

## 3. The Swedish Context

### 3.1. Land Use Planning

Land use planning and land development in Sweden is mainly regulated by the Planning and Building Act (plan- och bygglagen) and the Environmental Code (miljöbalken) that protects ecologically and/or culturally important land. Municipalities in Sweden decide how, when and where land development takes place in their geographical area, which is often described as a planning monopoly (see, e.g., [60]). They have several planning instruments which, according to the Planning and Building Act [61], are intended to promote urban development that creates equal, good and sustainable living environments. From this, we might conclude that human dignity, robustness and sustainability are central public values in urban planning and development in Sweden (c.f. [50]). Formal planning instruments include comprehensive plans (översiktsplaner) with guidelines for development in the municipality, legally binding detailed development plans (detaljplaner) used to regulate individual development projects and legally binding building permits (bygglov) required for the construction of new buildings. Whether land is publicly or privately owned, detailed municipal planning in the urban development process typically increases the economic value of the property, as seen both in Sweden [62] and other countries with similar systems (e.g., [41]), although this ultimately depends on the content of the planning regulations. The detailed development plans regulate the permitted uses of buildings and their size, sometimes including the size of individual dwellings [5]. Municipalities can choose the degree of detail included in their detailed development plans, although they are not meant to be more detailed than is necessary in relation to the plan's purpose.

### 3.2. Public Land Development and Municipal Land Allocations

Land development projects in Sweden can be implemented in different ways depending on land acquisition and land ownership [63]. Here, the focus is specifically on projects where the land is owned by a municipality, which has certain implications for land development. Many Swedish municipalities, especially those that are more populated, own significant portions of land (see, e.g., Table 2) and are therefore considered important suppliers of buildable land by property developers [22]. In 2020, 97% of all Swedish municipalities owned land they considered suitable for housing development, and 64% were planning on buying up more land for housing [64].

**Table 2.** Total land area and municipally owned land[1] in the largest Swedish municipalities by population.

| Largest Swedish Municipalities (by Population) | Total Land (ha) [1] | Municipal Land (ha) [1] |
|---|---|---|
| Stockholm | 18,716 | 10,163 |
| Göteborg | 44,788 | 23,530 |
| Malmö | 15,660 | 7621 |

[1] The data is from the most recent national survey on land ownership carried out in 2015 by Statistics Sweden.

Although the planning monopoly provides municipalities in Sweden with many opportunities to govern urban development, owning land offers them additional opportunities to steer urban development in individual housing development projects [5,65]. In public land development, municipalities use land allocations, land allocation agreements and final development agreements to steer development. There are three land allocation methods municipalities use to choose housing developers. These are concept competitions and price competitions (markanvisningstävlingar), and direct allocations (direktanvisningar), which is the most common method chosen motivated by the presumption that it requires less municipal resources [66,67].

Municipalities assign their chosen developers to municipal land using land allocation agreements, which they sign either during or after detailed planning. In the Swedish

Planning and Building Act [61], land allocation agreements are defined as "an agreement between a municipality and a developer (byggherre) that gives the developer the sole right to negotiate with the municipality for a limited time and under given conditions on the transfer or lease of a certain piece of land owned by the municipality for development" (author's translation). Once land allocation agreements have been signed, the developers work with the municipalities in an inter-dependency based relationship to produce final development agreements [22]. In most cases they also actively contribute to producing detailed development plans to coordinate land use planning with subsequent construction [22,62]. After this, the building plots are transferred to the developers, building permits are applied for and issued and building development can start [65].

In developments on municipally owned land, municipalities in Sweden capture value through the sale of the land and through various contributions included in municipal land allocation agreements and subsequent development agreements. When negotiated along with the detailed planning process, these contributions become a form of NDOs. In addition to this, developers may be charged by municipalities to cover part of the costs for technical infrastructure, such as roads, although this may not be extended to social infrastructure in Sweden. Valtonen et al. [5] found that, as a result of municipalities' rights for compulsory purchase, public land development in Sweden appears to be more efficient for public value capture, which is internalized after land acquisition, than private land development. Regarding N-NDOs, developers in Sweden are required to bear the administrative costs of preparing the detailed development plans and issuing building permits [40].

*3.3. Using Public Land and Negotiable Developer Obligations to Drive Sustainable Development and Innovation*

Municipal land allocations have become a central instrument for municipalities in Sweden to achieve sustainability-related public objectives in urban development, which are substantive public values. Municipalities in Sweden drive sustainable development and innovation through the use of sustainability criteria for land allocation and sustainability requirements in land allocation agreements that go beyond the national building regulations either in scope or in their content [8,9,65]. In sustainability-profiled districts, these types of sustainability requirements may entail, for example, the adoption of certain environmentally sustainable technologies and materials such as timber, energy performance, or calculating the environmental impact of various construction materials in order to perform life cycle assessments (LCAs) [65]. These contributions are initially agreed upon in return for the transfer of land, but are negotiable up until a final development agreement is signed. They are negotiated together with other land-use regulations for building permits, and are often also negotiated in relation to regulations in detailed planning, meaning they can be interpreted as a form of NDOs (c.f. [6]). The use of such sustainability-oriented NDOs is particularly prevalent in sustainability-profiled district developments, which are considered important flagship projects that lead sustainable development in many Swedish municipalities. These district developments exemplify how public–private partnerships are used in housing development to contribute to the advancement of sustainable practices (c.f. [68]) and to create public value (c.f. [54]).

There is some controversy over the use of municipal requirements that go beyond the current national building regulations (see, e.g., [69]). Since 2015, the Planning and Building Act has restricted municipalities from placing their own requirements on construction works' technical properties, which has been perceived as a major setback for municipalities' ability to drive sustainable development [65]. Requirements on construction works' technical properties that go beyond those stipulated by the national building regulations (see Boverkets byggregler) are typically referred to as special requirements (särkrav). The 2015 legal block was intended to improve conditions for meeting Sweden's growing demand for housing by reducing construction costs, following the logic of classic private property ideology [37]. The use of special requirements in cases when municipalities are acting as land owners has, however, continued despite the 2015 legal block [65,69]. It is uncertain

whether this is due to misinterpretations of the law or deliberate transgressions, but either way there is some observed uncertainty and confusion regarding the types of requirements that municipalities can legally place on building development [65].

Högström et al. [70] argue that translating sustainability objectives into various requirements is an important part of the planning process that links individual development projects to municipal strategies. Following Crow's [42] suggestion, both the product and the process of using sustainability criteria and sustainability requirements in practice are investigated herein. Sustainability criteria used to choose developers in municipal land allocation competitions and sustainability requirements in municipal land allocation agreements are here interpreted as potential NDOs. This implies that Swedish municipalities are using both macro value capture instruments and indirect instruments within the same projects [37]. The content of these sustainability criteria and requirements is evaluated in terms of public value outcomes using Benington's [14] four dimensions of public value.

## 4. Materials and Methods

### 4.1. Case Selection

Case studies were used to gather in-depth and context dependent knowledge [71] of how NDOs and public land development are utilized in sustainability-profiled districts in order to answer the first research question. These empirical case studies are used to discover and to test what is referred to as "tools of explanation" [72] (p. 515), which both shape and are shaped by theory. These same cases were also used to exemplify the types of public value outcomes that Swedish municipalities currently pursue using these public value capture instruments in order to answer the second research question. Analyzing a small number of empirical cases, also referred to as cross-case analysis, makes it possible to identify significant differences between them [72]. However, the potential for generalizations is consequently limited. The multiple case study was carried out using an abductive approach, which entails a simultaneous or iterative process of carrying out the empirical fieldwork, analyzing material and consulting literature [73]. Eisenhardt [74] vouches for this approach to case study research that seeks to create new knowledge, as opposed to confirming existing theory.

For the study, five ongoing sustainability-profiled district developments were selected from different Swedish municipalities. Although they are becoming more common, there is still a very limited number of such ongoing district developments in Sweden. Therefore, in order to "maximize the utility of information" from a limited number of cases, the districts were chosen based on expectations of the information that could be gathered from them, which Flyvbjerg [71] (p. 230) refers to as information-oriented selection. They were mainly selected based on the municipalities' very high ambitions on sustainable development and innovation and the considerable use of sustainability criteria in municipal land allocations and requirements in development agreements. Stake [75] advocates for the study of extreme and unusual cases, such as these, because they can reveal things that have previously been overlooked in typical cases. According to Flyvbjerg [71] (p. 13), extreme cases also "activate more actors and more basic mechanisms" and thereby generate more information.

The developments were found through the Swedish government's network for new city districts 2020 report, which presents a list of ongoing and upcoming urban development projects "with especially high ambitions concerning sustainability and innovation" [76] (p. 9). Stockholm Royal Seaport (SRS), Älvstaden (ÄS), Västerport (VP) and Barkarbystaden (BS) are all in the report. Hyllie (HL) is not in the report but was identified through contacts from Nyhamnen, a sustainability-profiled district near completion in Malmö that is in the report. The districts are located in the three largest municipalities (by population size) and two smaller municipalities in different parts of Sweden (see Figure 2). SRS, ÄS and VP are all located in city centers by the water in old industrial port areas, making them waterfront brownfield developments. BS and HL, on the other hand, are inland developments located in the urban periphery of large cities with good connections to city centers. HL is being developed on what was previously agricultural land and BS is largely being developed on

an old airfield, making much of these districts greenfield developments. These differences provide a basis for the cross-case analysis [74].

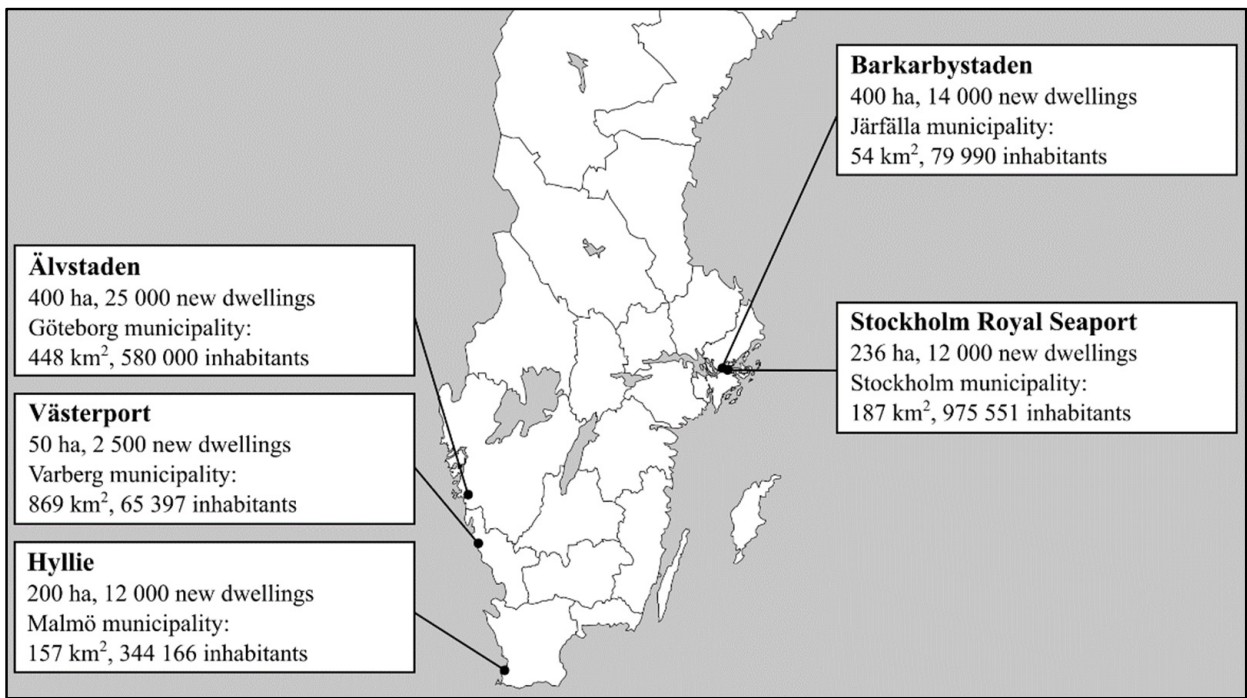

**Figure 2.** Case studies.

### 4.2. Collection of Materials

Material collected from each case consists of available public documents from public records and semi-structured interviews with at least one municipal project manager from each case (see Table 3), using data triangulation to improve the validity of the results [74]. The interviews for SRS were carried out first between November 2018 and September 2019. Some municipal planning project managers were interviewed twice resulting in the long data collection period. The interviews for the other cases were then carried out between November 2020 and March 2021 (over zoom due to Covid-19 restrictions). The interviews were all recorded and transcribed by the author. Themes explored in all interviews included: what public value the municipality sought to create in the district development as a whole and within individual stages, and how they use criteria in land allocation competitions and requirements in municipal land allocation agreements to achieve their public value creation objectives.

Documents from the cases include; sustainability programs outlining the visions for the districts, invitations to land allocation competitions for individual stages containing the sustainability criteria used to choose developers, sustainability/quality programs with project-specific sustainability requirements for the various stages included as attachments in the land allocation agreements signed with housing developers, and detailed development plans. Other documents, such as sustainability reports, were also used to gain a richer and more holistic understanding of each case. The documents comprise the primary data used to analyze the actual content of the municipalities' sustainability-oriented NDOs in order to answer the second research question.

**Table 3.** Summary of material for each case.

| Case | Documents | Interviews (60–120 min/Interviewee) |
|---|---|---|
| Stockholm Royal Seaport (SRS) in Stockholm | -5 sustainability programs with requirements for land allocation competitions and agreements for different stages<br>-Mobility index with additional sustainability requirements related to transportation<br>-4 invitations for land allocation competitions for different stages<br>-Draft for land allocation agreements<br>-Sustainability program for all of SRS<br>-9 sustainability reports<br>-Sustainability requirements for all construction on municipal land | 3 municipal planning project managers (2 were interviewed twice) (PM SRS)<br>1 municipal sustainability strategist (SS SRS)<br>1 municipal consultant responsible for sustainability coordination<br>1 municipal contract lawyer |
| Älvstaden (ÄS) in Göteborg | -3 sustainability programs with land allocation requirements for different stages<br>-2 invitations for land allocation competitions for different stages<br>-Vision document for all of Älvstaden<br>-2 sustainability reports | 3 sustainability process leaders from different stages (PL ÄS) |
| Västerport (VP) in Varberg | -Vision document for all of Västerport<br>-Sustainability program for all of Västerport<br>-Invitation and program for a land allocation competition in the first stage | 3 municipal planning project managers (PM VP) |
| Hyllie (HL) in Malmö | -A climate contract for all of Hyllie<br>-Environmental program for all of Hyllie with land allocation requirements<br>-Invitation for land allocation competition<br>-Land allocation evaluation<br>-Land allocation program<br>-9 housing projects summaries (including a breakdown of specific measures implemented from the environmental program)<br>-4 detailed plans | 1 municipal planning project manager (PM HL) |
| Barkarbystaden (BS) in Järfälla | -Program for all of Barkarbystaden<br>-Invitation and program for a land allocation competition<br>-4 quality programs with land allocation requirements for different stages (attachments in detailed plans) | 1 municipal planning project manager (PM BS) |

To improve validity, as much available material as possible was gathered to gain a sufficiently rich understanding of each case and its context before comparing them [74,77]. The amount of available material, however, varied somewhat between the cases for a few reasons. Firstly, the objective of the multiple case study was to explore similar phenomena in different empirical contexts, notably in different municipalities. There is great variation between Swedish municipalities when it comes to both their organizational structures and how they decide to structure their planning processes [78]. As a result, comparable roles and responsibilities are divided between different numbers of individuals with different titles. In addition to this, more interviews were carried out in the SRS case because it was first investigated in more depth, as a form of pilot study, to identify relevant points of inquiry before gathering more relevant material from other cases. In abductive case study research it is advisable to adjust data collection as more knowledge is acquired [74]. As a result of keeping the later interviews more focused on relevant themes, and carrying out longer interviews in cases where more roles and responsibilities were placed on fewer individuals, the amount of relevant material varies less than the number of interviewees for each case would suggest. This helped ensure that the interview material gathered from the different cases was comparable in relation to relevant content.

*4.3. Analysis*

A thematic document analysis [79] was carried out using Benington's [14] four dimensions of public value. These dimensions were applied as a typology to categorize public sustainability objectives and their corresponding criteria and requirements found in documents and discussed in the interviews. The public sustainability objectives found in various documents are all described in terms of desired public value creation. This made it fairly straightforward to divide them into the four dimensions of public value which were treated as main themes in the analysis process. These public sustainability objectives are broken down into more specific sustainability criteria and requirements for the different stages of the district developments, which each consist of several housing development projects carried out by different housing developers. Criteria and requirements were gathered in an excel file and divided according to the respective public sustainability objectives, and those were divided into the four public value dimensions. The various requirements were then reviewed according to the definition for each public value dimension to ensure that they were appropriately categorized. Several requirements were found to fit more than one public value dimension and were therefore categorized as belonging to more than one theme.

After compiling and categorizing criteria and requirements from the cases, the analysis consisted of iteratively interpreting each case using the theory on public values and public value creation and searching for cross-case patterns of similarities and significant differences [72,79]. Notable differences between the cases' requirements were found within each public value dimension, which resulted in the formation of categories within each theme. Comparing the cases, as well as comparing the results from the document analysis with material from interviews, aided the identification of relevant patterns and formation of categories. Differences between the cases also enabled a cross-case search for patterns based on the dimensions and categories [75]: size of the municipality, location of the districts in relation to city centers, waterfront vs. inland development, and brownfield vs. greenfield development. The results were presented to and discussed with academic peers to address bias through member-checking.

## 5. Results

*5.1. Using Public Land and Negotiable Developer Obligations in Sustainability-Profiled Districts to Create Public Value*

Swedish municipalities are increasingly using the allocation of municipal land to develop individual urban districts with profiles related to environmental sustainability or sustainability in general, exemplified by all of the cases in this study. These sustainability-profiles are sometimes also combined with other buzzwords such as 'smart', denoting ambitions to develop technological infrastructure, as in the HL case. This has been a growing trend for over two decades since the success story of Hammarby Sjöstad in Stockholm, a sustainability-oriented urban district development that started in the 1990s. Although they are not the norm, several of the larger Swedish municipalities are now developing at least one high-profile sustainability-oriented district at a time to drive sustainable development and showcase their efforts and ambitions to achieve sustainability-related public values. The districts in this study function as testbeds for both new planning policies and practices as well as new sustainable construction solutions and practices. They receive many resources from municipalities as they are expected to generate societal value in the form of new sustainable practices and solutions that can eventually be adopted in mainstream urban development and construction. Resources are recovered through land prices which are especially high in the waterfront districts SRS and ÄS located in large city centers.

> *"There are a lot more resources [from the municipality] here compared to normal projects . . . which is connected to the high land prices. We justify this by carrying out projects that benefit the whole city"* (PM SRS)

The municipalities in all of the cases dedicate a large part of their efforts and resources to developing the district profiles consisting of contextually specific visions and objectives,

which consequently also produces attractive conditions for developers. The sustainability-related profiles typically generate a lot of public interest and positive publicity. When combined with attractive locations, such as areas by the water in or near the city center (e.g., SRS, ÄS and VP), the municipalities are able to ask for high land prices. They are legally obligated to sell the land to developers for the price that the market currently values it at, according to EU state aid rules, although municipalities that own almost all developable land for housing are essentially market makers. While it does happen that municipalities sell their land under market value, this is not the case in the sustainability-profiled districts investigated herein.

The municipalities allocate their land in the districts using concept competitions, direct allocations and sometimes price competitions. The most common method used in all five cases is to invite property developers to compete for land based on specific sustainability criteria with a fixed price, which the two smaller municipalities (VP and BS) see as quite a distinct practice for these types of districts. In these instances, the developers compete by either providing a proposal for their solutions and designs or referencing previous projects.

> *"Västerport is the first project in Varberg municipality that has a sustainability program . . . It has permeated much of what we do. When we allocated land to the 12 developers they were evaluated on sustainability and design with a fixed price. We have not done that here before . . . Each area had 13 criteria for design and 13 for sustainability"* (PM VP)

In SRS, ÄS and BS, land allocation competitions are also combined with direct land allocations where housing developers are directly chosen by the municipality without an invitation for other developers to compete. For instance, in BS they directly allocate land to housing developers that have taken part in previous land allocation competitions and have presented proposals that gained favor with the municipality.

> *"In the field alone there are approximately 140 blocks that are to be built, so we do not have the energy to carry out actual competitions for all of the blocks and a few that have good ideas get direct land allocations too"* (PM BS)

In SRS, they have also allocated land where "the sustainability requirements in the program were a basic precondition and the developers compete over price" (SS SRS). A similar option, adopted in other sustainability-profiled district developments in Sweden not directly studied herein, is to have the developers compete for land based on price and then receive discounts for implementing various sustainability measures.

Sustainability criteria for choosing developers in municipal land allocation competitions, and subsequent requirements in land allocation agreements, are used in all of the cases to challenge developers to innovate and create various forms of public value. They are outlined in some form of sustainability programs included as attachments in the land allocation agreements, which can be "called different things but are usually some form of document with checklists and illustrations that say something about what and how we will build" (PL ÄS). The developers have to follow these sustainability programs in order to be allocated land. After a developer wins and signs a land allocation agreement with the municipality, these project-specific sustainability requirements are negotiated until a final development right has been signed, typically alongside detailed planning. During this process, the developers often meet various challenges (see [80]), which result in alterations to the requirements and other land-use regulations through continuous dialogue with the municipality. This process of developing requirements can be interpreted as the municipalities translating public values and framing their desired public value outcomes.

Project-specific sustainability requirements from land allocation agreements can be seen as a form of NDOs when negotiated alongside detailed planning and building permits. However, the distinction between leveraging the transfer of land versus specific land-use regulations is often not explicit. Their purpose is also not directly tied to cost recovery or traditional infrastructure provision. They entail implementing solutions that will provide added public value in a more general form of contributions to the public sphere as well as enabling innovation and improvement in the municipalities' own practices and future

policies. Many of the requirements in the cases exceed current national building regulations and cannot be included in the detailed development plans according to the Planning and Building Act chapter 8 §4a [61]. Municipalities thereby regulate more in public land development, through the use of NDOs in development agreements, than in developments on private land.

> *"The municipality owns almost all of the land, which has made it easier to place requirements. When the municipal special requirements legislation came into force it was easy to get a feeling of having our hands tied, but when the municipality owns the land themselves you can still play around a bit more with certain aspects. It is an advantage that the municipality is such a large landowner . . . around the stations there is only municipal land which we have planned and been able to sell on our own terms to different actors"* (PM HL)

The sustainability-oriented NDOs in these cases are indirect instruments as they are not explicitly intended to capture unearned land value increments. It is unclear what exactly the municipalities are leveraging in exchange for these sustainability-oriented developer obligations, which allocate the responsibility of realizing various public sustainability objectives to the housing developers. For the housing developers, this often entails some form of innovation and fitting the additional work with subsequent costs and added risk into their own project budgets (see [80]). If there are no unearned land and building value increments that are not reflected in the land price, the municipalities might instead be capturing part of the developers' profits. The developers' profits are generally high enough to motivate them to participate anyway, although this is highly dependent on the housing market. Then, there are also some identifiable benefits that the municipalities are leveraging more openly but informally. Of particular note is how housing developers profit from the marketing potential of these projects in the long-term. They use them as showcase projects to demonstrate their innovation and sustainable construction capabilities to both the market and to municipalities for future land allocations. There is an understanding that delivering successful projects in sustainability-profiled districts may lead to more favorable land allocations from the same municipality in the future, which municipal planners sometimes refer to as trust capital (förtroendekapital). This could explain why there is a significant interest from developers in Sweden to take part in these sustainability-profiled districts despite the risk of additional costs.

*5.2. Public Value Outcomes Municipalities Pursue Using Sustainability-Oriented Negotiable Developer Obligations*

All four dimensions of public value are discernable in all of the documents outlining the municipalities' visions for the districts, which are broken down into specific objectives. The objectives in these visions represent public values the municipalities seek to achieve. They translate these objectives into more general criteria for land allocation competitions and longer lists of more specific requirements. The sustainability requirements are included in attachments to the land allocations agreements and negotiated alongside other land-use regulations to produce a final development agreement. These NDOs typically differ somewhat between stages of the district developments, but are for the most part the same for housing development projects within the same stage: "For each new stage we try to continue working with aspects that have been good but also drive our work forward and have new requirements" (PM SRS). The individual stages are thereby used as testbeds for a specific set of requirements, which are sometimes focused on one public value dimension (typically ecological or social and cultural public value) more than the others to prevent inherent conflicts between them.

Although all public value dimensions are identifiable in the municipalities' objectives, they are not all necessarily achieved using NDOs. It should also be stressed that the prevalence of certain requirements in municipal land allocation agreements does not ensure that the associated public value creation objectives are actually achieved. There is still the possibility that the property developers are not able to implement the requirements in their

projects and/or that they are negotiated away for some reason as the projects progress. In other words, the NDOs investigated here are specific aspects of the municipalities' public value creation objectives and not guaranteed outcomes. The sustainability-oriented NDOs from the cases are categorized into the four dimensions of public value. This is summarized in Table 4, followed by a discussion of each dimension. This is not an extensive list of NDOs used in the cases, but an overview of those specifically related to sustainable development. A separate section is also dedicated to discussing objectives, criteria and requirements on innovation, learning and sharing knowledge, which typically span all or some of the different public value dimensions.

**Table 4.** Summary of sustainability-oriented developer obligations by public value dimension.

| Public Value Dimension | NDO Categories | NDO Examples | Prevalence in Cases |
|---|---|---|---|
| Ecological | Sustainable transportation | Bicycle parking, charging stations, mobility plan/index, limited parking for cars | SRS, ÄS, VP, HL, BS |
| | Waste management and recycling | Reduced waste during construction, recycling systems | SRS, ÄS, VP, HL, BS |
| | Energy efficiency and clean energy | Recycling heat from greywater, solar panels, sustainable and locally produced energy, insulation | SRS, ÄS, VP, HL, BS |
| | Sustainable and safe materials | LCA, recycled materials, regulating harmful chemicals | SRS, ÄS, VP, HL, BS |
| | Green areas/vegetation | Green area factor, green roofs, biodiversity | SRS, ÄS, VP, HL, BS |
| | Resilience and safe construction | Storm water management, assembly methods, material choices | SRS, ÄS, VP, HL, BS |
| | Environmental certifications | Certify buildings using a recognized certification system (e.g., Miljöbyggnad, Svanen, BREEAM, LEED) | ÄS, BS |
| Social and cultural | Mixed neighborhoods (equality in access to housing, integration) | Mixed housing forms (e.g., tenant-ownership apartments, rental apartments, student housing, elderly homes), affordable housing | SRS, ÄS, VP, HL, BS |
| | Shared (green) spaces | Courtyards, spaces for urban cultivation | SRS, ÄS, VP, HL, BS |
| | Vibrant/active streets and access to services | Flexible bottom floor premises | SRS, ÄS, VP, HL, BS |
| | Individual and community well-being | Lighting, daylight, quality of indoor environment (e.g., noise, harmful chemicals), safety considerations | SRS, ÄS, VP, HL, BS |
| | Design | Mixed façade designs, shared design aspects, type and quality of materials, art and decoration | SRS, ÄS, VP, HL, BS |
| Political | Stakeholder dialogues | Engage in dialogues with various stakeholders | ÄS, VP |
| Economic | Stimulate and attract sustainable business and support small businesses | Flexible bottom floor premises | SRS, ÄS, VP, HL, BS |

### 5.2.1. Ecological Public Value

Many of the public sustainability objectives from the cases directly align with the definition of ecological public value. Each district has some variation of one or several objectives to promote the development of environmentally sustainable solutions and practices and reduce negative impacts on the environment. Locating districts by the water versus inland results in the most significant differences regarding ecological public value creation objectives. In the waterfront developments SRS, ÄS and VP, a significant portion of their objectives concern the ocean, including, for example, mitigating pollution of the waters caused by dirty storm water and developing preventative measures for potential sea level rises. These are naturally not major concerns in the HL and BS cases located inland.

Despite differences in objectives, criteria and requirements for creating ecological public value in housing developments fall under very similar categories in each case.

All cases have requirements pertaining to sustainable transport, waste management and recycling, energy efficiency and clean energy, sustainable and safe materials, green areas and resilience and safe construction. Many of these requirements focus on the technical properties of the buildings and other amenities, meaning they can be measured. This makes formulating and following up on these requirements more straight forward than other types of requirements, such as those related to social public value. This is partly why they are also the most prevalent criteria and requirements among all cases presented herein.

The only category not present in all cases is environmental certifications. It is only the municipalities in the ÄS and BS cases that require developers to certify their buildings using some form of certification system that is recognized in Sweden. These certifications come with their own set of requirements and are quite common and popular in Sweden, although they are not free from criticism. For instance, the requirements are not always well suited for different geographical contexts, such as installing solar panels on buildings in the northern parts of Sweden where there is very little sunlight for half of the year. Although they are not required by all municipalities in this study, all of the municipalities do base several of their requirements on specific certification systems (most commonly Miljöbyggnad), referring to them in their sustainability programs.

### 5.2.2. Social and Cultural Public Value

All cases have social and cultural public value creation objectives. The subsequent criteria and requirements can also be grouped into categories that are identifiable in all of the cases. The ways in which these requirements are connected to social and cultural public value in the sustainability programs, however, differs significantly between the cases. The degree to which this connection is emphasized also differs significantly. For instance, in several cases, this connection is only made by interviewees as the actual requirements are not presented together with other sustainability requirements. One example is criteria and requirements related to design and urban form, considered as an important part of creating aesthetically attractive neighborhoods by interviewees in all cases, but only fully integrated with other sustainability-related criteria and requirements in ÄS and BS. Many of the criteria and requirements related to social and cultural public value are more open for interpretation than those that fall under ecological public value. Social sustainability, which is closely related to social and cultural public value, is currently recognized as something that needs to be developed further in these types of districts.

Creating mixed neighborhoods is an objective connected to creating social and cultural public value seen in all of the cases. For example, one of the main focus areas in the VP case is "variation for integration . . . [which includes] mixed forms of housing"(PM VP). The municipalities attempt to achieve these mixed neighborhoods by including different forms of housing other than the most common tenant-ownership apartments (bostadsrätter), such as student housing, elderly homes and rental apartments. This is used to try to minimize and tackle issues of segregation, which is a particularly big concern in the districts located in big cities. However, mixing different forms of housing within a neighborhood does not necessarily mitigate all forms of segregation (c.f. [81]). An especially common issue in these cases is that apartments become too expensive as a result of the high land prices and costs for the various sustainability-related NDOs.

> *"How do we ensure that this does not reinforce the segregation that already exists in Gothenburg? This is one of the big questions. How do we get affordable housing?"* (PL ÄS)

While there are more NDOs focused on creating ecological public value in these cases, from the interviews, it is clear that the municipalities are equally focused on creating social and cultural public value. Many of the municipal objectives that are connected to social and cultural public value are, however, more clearly achieved through land-use planning at the district or neighborhood level, rather than within individual housing development projects. One common example is creating a mixture of building façade designs.

> *"The land allocations were used to attain variation in buildings and housing types. You could say that we have 6 blocks, but we divided the block into smaller lots to get several developers in the same block to get this variation in material choices and heights and design to mimic a city that gets to grow over time"* (PM VP)

Another example discussed in all cases, although notably more central in SRS, ÄS and VP located in city centers, is creating physical and meaningful connections between the new district and other parts of the city. As with the ecological public value creation goals, there is a notable difference here between the brownfield waterfront developments in city centers (SRS, ÄS and VP) and the greenfield inland developments in the urban periphery (HL and BS). The waterfront developments in the city centers are treated as important symbolic cultural reflections of the city.

### 5.2.3. Political Public Value

In the vision documents, political public value objectives are either separate or grouped together with social public value objectives. The municipalities in all of the cases have some form of objective to engage citizens and promote dialogue. However, the municipalities consider achieving political public value outcomes through stimulating and supporting more citizen dialogues and public participation during earlier stages of the planning process to be their responsibility. The municipalities in SRS, ÄS and VP dedicated significant resources and efforts to involve citizen groups, and the local population in general, in generating the overall vision for the districts before carrying out detailed planning. This was not as prominent in HL and BS. Again, there is a clear divide between the waterfront city center districts and the inland districts in the urban periphery. Aspects of the district visions are translated into NDOs. Political public value is thereby created inversely using NDOs.

> *"We have set a vision and goals to be broken down . . . translated into concrete action . . . tied up through agreements and in detailed plans . . . The vision was first developed through dialogues with 3 000 citizens . . . It was well rooted in the politics, from both right to left"* (PL ÄS)

The political public value objectives are broken down into very few, and in most cases no, NDOs. Only ÄS and VP explicitly require the developers to engage in dialogue with stakeholders, such as citizens and businesses, during their housing development planning. Overall, there is more citizen engagement in ÄS, VP and SRS than in HL and BS, which is a result of developing districts in city centers where there are many affected inhabitants in the surrounding areas compared to undeveloped land in the urban periphery.

### 5.2.4. Economic Public Value

All of the cases have municipal objectives to generate different forms of economic activity and enterprises in the districts. They are typically looking to promote and support sustainable and small businesses in the district, as well as attract sustainable business from elsewhere. However, these objectives are generally broken down into very few NDOs. The most common set of requirements connected to economic public value are for the housing developers to include flexible premises for small business enterprises along streets on the ground floors of their buildings, seen in all of the cases. In all five cases, these requirements are presented under visions related to developing "attractive" and vibrant districts, termed "living" cities, which they relate to creating both economic and social and cultural public value. Requirements connected to creating ecological public value are typically applied to these premises as well.

> *"How do we get a mix of businesses so it is not only those that have the opportunity to pay high rents but a range of available businesses that support a sustainable lifestyle for those who will be living here? . . . We want to have sustainable growth, attract business and get more inhabitants to the city and give them a reason to settle here and start businesses"* (PL ÄS)

### 5.2.5. Improving Public Value Creation through Innovation, Collaboration and Learning

Objectives, criteria and requirements on innovation, learning, collaboration and sharing knowledge, specifically in relation to creating the different dimensions of public value, are present in all five cases. These NDOs are not intended to create any specific public value outcomes, but to improve public value creation processes. These are either presented separately and applied generally across the dimensions of public value or interwoven as criteria and requirements specifically focused on generating innovation and learning for various specified dimensions of public value. Examples include requiring developers to participate in collaborative competence building and coordination activities, as well as collaborating on shared facilities. Municipal planning project managers from several of the cases explain that desired innovation from housing developers includes both developing new sustainable solutions and practices and finding ways to make them economically feasible and ideally lucrative and/or desirable to implement in housing development projects. This is an important contribution from the housing developers because it will determine whether new sustainable solutions and practices have any chance of being adopted in the construction industry at large.

> *"Implementing these requirements obviously costs more, but it is also a societal benefit. Pilot projects are like that. You see afterwards what you can get out of it"* (PM SRS)

While there is a strong emphasis on learning and collaboration within all of the districts, several of the interviewees recognize that learning between districts and between districts and mainstream urban development remains fairly underdeveloped and underutilized.

## 6. Discussion

In the cases presented herein, Swedish municipalities are allocating responsibilities for financing and creating public value, in the form of sustainable development and innovation, to developers using NDOs. These sustainability-oriented NDOs are included in development agreements as a part of the public land development process, answering the first research question. Developer obligations and active land management and development are both seen as public value capture instruments [18,37]. However, it is unclear what value increases the sustainability-oriented NDOs are capturing that are not already reflected in the land prices, which may be an issue for equitable value capture (c.f. [5]). There may be some added marketing potential for developers that the municipalities are creating from profiling the districts that they can then leverage, but this is not made explicit. Although ambiguity regarding motivating rationales for indirect instruments is common, a lack of transparency can lead to political and legal challenges [7,37]. The findings do, however, illustrate that sustainable development is being used as a motivating rationale for NDOs (c.f. [5,6,18,37]).

Developer obligations are typically considered a public value capture instrument used to finance public urban infrastructure provision (e.g., [6,7,17,18]). However, it is questionable whether all of the housing developers' contributions in the sustainability-profiled district developments from this study can be considered public urban infrastructure. In some cases, they indirectly contribute to the sustainable development of public urban infrastructure provision, but do not necessarily entail directly financing or providing it. The public value outcomes for society that the municipalities pursue in these districts go beyond the typical public urban infrastructure discussed in relation to developer obligations (e.g., [6,7]) and public sustainability objectives identified by Valtonen et al. [5]. On the other hand, the connection between many of these sustainability-oriented NDOs and the developments are arguably clearer than for some off-site social infrastructure sometimes financed using developer obligations (c.f. [7]).

Heeres et al. [45] argue that the value of value capturing in land use planning goes beyond financial value as it also enhances cooperation between fragmented actors, which they consider as cooperative value. The findings here illustrate that NDOs can also create different forms of public value to achieve a larger range of public values. In sustainability-profiled

district developments in Sweden, municipalities seek to create what Benington [13,14] refers to as the ecological, political, social and cultural, and economic dimensions of public value, answering the second research question. These public value outcomes are the desired products of their sustainability-oriented NDOs (c.f. [42]). In this study, all of the most notable differences regarding public value creation objectives and sustainability-oriented NDOs are between brownfield waterfront districts located in city centers and greenfield inland districts located in the urban periphery, and surprisingly not between large and small municipalities. For example, the brownfield waterfront districts in city centers focus much more on issues related to the ocean and citizen engagement.

The findings illustrate that NDOs intended to create ecological public value and social and cultural public value are currently more common in Swedish sustainability-profiled districts than NDOs for creating political and economic public value. This raises the question of which public values municipalities should be prioritizing, which is a central question within the public values literature [48]. The findings also illustrate how the NDOs for the different dimensions of public value differ in regard to the type of public values they are achieving. For instance, NDOs creating ecological, social and cultural and economic public value are arguably more closely related to achieving substantive public values, while NDOs creating political public value are more closely related to achieving procedural public values (c.f. [51,52]). From this, we might conclude that NDOs are more suitable for achieving substantive public values than procedural public values which are not outsourced in the same way. Then, there are also NDOs that fit within the ecological public value dimension, such as those concerning energy efficiency, which could be connected to performance public values as well. An argument can also be made for NDOs that focus on creating public value through the innovation and improvement of public value creation processes being closely related to achieving performance public values. NDOs creating ecological public value are also much more technical, and thereby easier to measure compared to the NDOs within the other public value dimensions, which are much more subjective. This means that public value can cover the range from the objective (c.f. [46,49]) to the more subjective (c.f. [53]), within the same development project.

Although the argument laid out here calls for a broader conception of public value to understand the desired outcomes of these sustainability-oriented NDOs, some caution is warranted to avoid undesirable confusion and concept creep. Here, this was achieved by distinguishing between public revenue and the four dimensions of public value [13,14]. Creating and capturing different dimensions of public value also exacerbates the difficulty of trying to determine who should be capturing what in property development (c.f. [38]). The municipalities are creating value that the property developers benefit from, such as development rights and marketing potential. However, the value that they capture comes in a myriad of different forms of public value, ranging from public revenue to innovative wastewater recycling systems, mixed neighborhoods and the potential development of future policymaking. It is not clear how these different forms of value should be compared in an equitable or practical manner. Currently, these issues are resolved in negotiations as developers evaluate what is technically and financially feasible to implement. However, it is not clear how they weigh this against the future benefits of having demonstration projects that can be used to win more desirable land allocations, among other things.

Similarly to those in other countries (e.g., [5,21,23]), municipalities in Sweden are limited in the contributions they can require from developers. However, when they own land, there seems to be more leeway for NDOs to be included in development agreements. However, this does also result in more political and legal concerns as the limitations are somewhat unclear (c.f. [65,69]), which is already a notable issue with regard to NDOs [7]. NDOs can, on the other hand, be modified to suit the local context of a development project (c.f. [17]), making them more flexible in comparison to for example sustainability-related certifications. Similarities between the cases presented here do, however, indicate that municipalities' sustainability objectives might run the risk of becoming generic, and thereby

less contextually embedded, as sustainability-profiled districts become more common and municipalities increasingly mimic each other.

## 7. Conclusions

### 7.1. Theoretical Contributions

Contributions have been made to the ongoing discussion about developer obligations as public value capture instruments (e.g., [6,16–18]) by exploring the current use of sustainability-oriented NDOs in Sweden. Contributions have also been made to the literature on public land development by presenting cases that illustrate its use in sustainability-profiled district developments, as well as presenting more cases from the Swedish context. Sustainability-profiled district developments in Sweden were presented as an example in which municipalities seek to create and capture different dimensions of public value using NDOs and public land development in order to achieve different public values. Our findings were based on five case studies of sustainability-profiled district developments in Sweden. The process of using sustainability-oriented NDOs in these cases was first explored from a public value capture perspective. The municipalities' desired outcomes were then identified and analyzed from a public value creation perspective adopted from the field of public management. The public value outcomes that these municipalities are currently pursuing were categorized using Benington's [13,14] four dimensions of public value as a typology. This analysis revealed that municipalities in Sweden use NDOs to pursue a wide range of public value outcomes that range from the more objective (c.f. [46,49]) to subjective (c.f. [53]). It is questionable whether all of the desired public value outcomes should be considered as contributions to public urban infrastructure. Differences regarding sustainability-oriented NDOs were mainly identified between brownfield waterfront districts in city centers and greenfield inland developments in the urban periphery.

### 7.2. Implications for Policy and Practice

The study presented herein illustrates how the use of sustainability-oriented NDOs differs between public value dimensions. Methods for capturing public value in public land development should be tailored for the different types of public value outcomes that are desired. NDOs are more suitable for some of these public value outcomes than others. For instance, NDOs oriented towards the creation of ecological public value are quite well established in Sweden. Although they have not come as far, NDOs oriented towards the creation of social and cultural and economic public value are also being developed in sustainability-profiled district developments. However, based on the findings, the utility of using NDOs to create political public value is questionable, especially considering as it is already well established in planning law.

As Francart et al. [65] previously suggested, the legal framework surrounding special requirements set by municipalities acting as land owners should be clarified to avoid legal confusion and political contention. Given the objective to be at the forefront of sustainable innovation and development in these districts and to develop new and innovative ways of working, focusing on establishing limitations, as suggested by Hendricks et al. [7], might seem somewhat counterproductive. It is, however, paramount that municipalities are at least transparent in their work. Learning and sharing knowledge between sustainability-profiled districts should also be improved, especially to help municipalities with less experience. Since sustainability-profiled districts are becoming more common in Sweden, there are more opportunities to learn and share knowledge regarding public value creation between them. Municipalities should, however, be careful not to completely copy each other to ensure that their developments are still being adapted to the local context.

### 7.3. Limitations and Future Studies

The study presented herein is limited to the Swedish context. Future studies could therefore investigate and compare the use of sustainability-oriented developer obligations in different national contexts. This study also focused more on the desired products of

sustainability-oriented NDOs, conceptualized as desired public value outcomes, and less on the process of actually capturing them. Negotiations that take place after land is allocated and up until there are final development rights typically take several years, and it is likely not possible to keep all initially desired aspects. Implementing these requirements might call for different processes to be developed. Future studies could therefore further explore the process of capturing these other forms of public value throughout the development process, and investigate whether desired public value outcomes are actually achieved.

Future studies should also continue exploring different conceptions of public value in relation to public value capture. The study presented herein illustrates that there is a need for more research investigating different forms of value in the context of public value capture. Other important questions for further investigation include who is determining what public value outcomes to pursue and how they are making these decisions in this particular context (c.f. [48]). It would also be worthwhile to investigate whether there are hierarchies among the desired public value outcomes (c.f. [48]), since these may influence how municipalities choose to prioritize public value creation objectives when they are in conflict with each other.

This study also focused on the process of using sustainability-oriented developer obligations together with public land development and the desired public value outcomes of this practice, but did not do much to connect this to financial elements such as public costs and revenue. Future research is needed to help bridge this gap between the findings presented here and the actual value capturing involved in using sustainability-oriented NDOs such as those in the sustainability-profiled districts investigated herein. Finally, in the study, it was found that municipalities include LCA as a requirement in municipal land allocations in sustainability-profiled districts. Since land development is an important part of environmental impact assessments, this relationship should be investigated further.

**Funding:** This research was funded by Svenska Forskningsrådet Formas; grant number 2016-20103.

**Data Availability Statement:** Materials may be retrieved from the author upon reasonable request.

**Conflicts of Interest:** The author declares no conflict of interest. The funders had no role in the design of the study; in the collection, analyses, or interpretation of data; in the writing of the manuscript, or in the decision to publish the results.

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
