# Peer review of "Using Sustainability-Oriented Developer Obligations and Public Land Development to Create Public Value"

_sustainability, doi:10.3390/su14010057_

Round 1
Reviewer 1 Report
In Introduction, the authors well present why we need to look into the role of the third sector developers and their public value creation in urban development, especially by emphasizing public land development matters in the global context, not just the Swedish case.
In 2.1. Public value capture instruments, the authors divide public land value instruments into direct and indirect, and insist that indirect measures are more pragmatic and flexible. It would be better if authors explicitly present some detailed examples of direct, indirect, and mixed cases using, for example, diagrams or pictures, even though the authors cite the previous literature.
In line 193, do you mean the “two” by “direct and indirect”?
For the first paragraph in 2.2. Public land developments and municipal land allocations, please provide a table that shows major municipals’ total land area and the area of publicly-owned land.
General comments: The paper well presents qualitative results in public land development (ambiguity and vagueness in sustainability-oriented NDOs clauses, public urban infrastructure delivery, and public value outcomes), and well discusses the implications for policy and practice in public land development, which can be summarized as that the combination of the public value creation tailored to each municipality’s context and the cooperation between municipalities is a recipe for the success of public land development. I think that these findings and implications well contribute to the existing literature on public land development and the public-private partnership.
Author Response
In Introduction, the authors well present why we need to look into the role of the third sector developers and their public value creation in urban development, especially by emphasizing public land development matters in the global context, not just the Swedish case.
Author response: Thank you for this positive comment.
In 2.1. Public value capture instruments, the authors divide public land value instruments into direct and indirect, and insist that indirect measures are more pragmatic and flexible. It would be better if authors explicitly present some detailed examples of direct, indirect, and mixed cases using, for example, diagrams or pictures, even though the authors cite the previous literature.
Author response: Thank you for this suggestion. I have included a diagram illustrating examples of direct, indirect and macro value capture instruments at the end of section 2.1.
In line 193, do you mean the “two” by “direct and indirect”?
Author response: Thank you for pointing out that this sentence was unclear. The “two” was not referring to direct and indirect instruments but specifically to developer obligations and public land development, which are both public value capture instruments. I have rephrased the sentence to make this clearer and have specified what type of public value capture instruments they are. According to Alterman [37], public land banking and development can be categorised as a macro public value capture instrument and developer obligations are typically indirect instruments.
For the first paragraph in 2.2. Public land developments and municipal land allocations, please provide a table that shows major municipals’ total land area and the area of publicly-owned land.
Author response: Thank you for making this suggestion. I have added a table in section 3.2. with the total land area and municipally owned land in the three largest Swedish municipalities (by population).
General comments: The paper well presents qualitative results in public land development (ambiguity and vagueness in sustainability-oriented NDOs clauses, public urban infrastructure delivery, and public value outcomes), and well discusses the implications for policy and practice in public land development, which can be summarized as that the combination of the public value creation tailored to each municipality’s context and the cooperation between municipalities is a recipe for the success of public land development. I think that these findings and implications well contribute to the existing literature on public land development and the public-private partnership.
Author response: Thank you for your encouraging words and your valuable feedback.
Reviewer 2 Report
The presented publication represents a quality scientific study focused on a very current and relevant topic. The article has a clear structure and the continuity of its individual parts is logical.Comments:
1. In 2. Theoretical framework; 2.2. Developer obligations - the NDO, N-NDO are clearly and relevantly presented and discussed,
however, in 3. Swedish context is this a very important insight missing e.g. N-NDO in the Swedish context at all.
2. In 4. Materials and methods; 4.1 Case selection - there is clearly explained the selection of 5 districts
(based on the municipalities’ high ambitions on sustainable development and innovation), however - all districts and case studies are seaside;
inland district is missing - and my question is: Can't this fundamentally affect the results of the analyzes and thus the overall findings of the study in question?
3. Table 2: Summary of material for each use - there are inequalities regarding interviews in different districts; e.g.
Stockholm Royal Seaport – 5; Hyllie in Malmo and Barkarbystaden in Järfälla – 1; there is a clear explanation here:
"More interviews were also carried out in the SRS case because this case was investigated first in more depth to identify relevant points of inquiry before gathering material from other cases"
however, my question is: Can these irregularities significantly skew the overall results?
4. Developer´s obligations and land development are the significant parts of the Environmental Impact Assessment (EIA) process. Is there such and(or) similar literature or case studies in Sweden dealing with these relationships? If so, the article would certainly benefit in context.
Author Response
The presented publication represents a quality scientific study focused on a very current and relevant topic. The article has a clear structure and the continuity of its individual parts is logical.
Author response: Thank you for your positive comments.
Comments:
1. In 2. Theoretical framework; 2.2. Developer obligations - the NDO, N-NDO are clearly and relevantly presented and discussed, however, in 3. Swedish context is this a very important insight missing e.g. N-NDO in the Swedish context at all.
Author response: Thank you for pointing out this shortcoming. The main N-NDO seen in the Swedish context has been presented at the end of section 3.2. To avoid making the paper too long, and since the focus of the study is on NDOs, this mention is kept short and concise. To avoid any confusion, I have also clarified this focus on NDOs more throughout the paper.
- In 4. Materials and methods; 4.1 Case selection - there is clearly explained the selection of 5 districts (based on the municipalities’ high ambitions on sustainable development and innovation), however - all districts and case studies are seaside; inland district is missing - and my question is: Can't this fundamentally affect the results of the analyzes and thus the overall findings of the study in question?
Author response: Representing both waterfront and inland district developments was taken into consideration when designing the study. Thank you for bringing it to my attention that this was not mentioned in the text. Three of the districts in the study are located by the water and two are inland. Figure 2 is perhaps misleading in this regard since all of the municipalities are located very close to the coast. Therefore, I have included some more details about each case in the text at the end of section 4.1. and also discuss these differences more throughout the results.
- Table 2: Summary of material for each use - there are inequalities regarding interviews in different districts; e.g. Stockholm Royal Seaport – 5; Hyllie in Malmo and Barkarbystaden in Järfälla – 1; there is a clear explanation here: "More interviews were also carried out in the SRS case because this case was investigated first in more depth to identify relevant points of inquiry before gathering material from other cases" however, my question is: Can these irregularities significantly skew the overall results?
Author response: Thank you for raising this concern. There are some inequalities regarding the number of interviewees for each case, although the amount of relevant content from each case is comparable, which is now explained more clearly in the second paragraph of section 4.2. I also highlight that the differences between the cases provide opportunities to investigate in-depth the use of sustainability-oriented negotiable developer obligations in different empirical contexts, albeit at the expense of perfectly controlled data collection. The risk of significantly skewed results is minimised by taking these differences between the cases into consideration more during the analysis, and ensuring that the analysis is based on a contextually rich understanding of each individual case.
- Developer´s obligations and land development are the significant parts of the Environmental Impact Assessment (EIA) process. Is there such and(or) similar literature or case studies in Sweden dealing with these relationships? If so, the article would certainly benefit in context.
Author response: Thank you for this suggestion. The most notable study that makes this connection between assessing environmental impact and developer obligations and land development in Sweden is Francart et al. [65]. They specifically highlight the prescription of life cycle assessment (LCA) as a sustainability requirement that municipalities include in land allocation agreements. If I have understood correctly, EIA and LCA processes are somewhat comparable. I have presented this on page 8 line 357-358. Since the same thing was observed in this study, LCA is included as an example of a developer obligation in Table 4 on page 15. To highlight this connection further, LCA has also been brought up as something for future research to investigate further (in section 7.3.).
Thank you for your perceptive comments and insightful suggestions.
Reviewer 3 Report
Thank you for the paper on this interesting topic of developer obligations and their contribution to public values. On forehand, I was very much interested because a more public value driven view on this kind of instruments is really needed. Unfortunately, the paper was in my view not able to really fill this gap. Firstly because of the limited theoretical work on public value and secondly because of the thin case study which led to a descriptive instead of a comparative analysis. I will elaborate these points.
Firstly, the work on public value. Public value is an important concept in this study. It is not necessary to involve all work on public value. However, I miss in theoretical part some important scholars and discussions. You limit your understanding of public value creation to the work of Moore. His work is important but also discussed, especially they way he presents public values as objective measurable values and professionals as the actor to measure these values. All this kind of discussions including the work of other scholars (e.g. Bozeman, Meynhardy, Stoker) is missing.
Secondly, the cases have the potential to be compared. However, the cases are only used to describe the way NDO’s could lead to public values. This could be interesting if also was analyzed whether this value was realized, but that is not analyzed. This could be interesting if more systematically was analyzed how the kind of NDO was related to the kind of public values, but that is not analyzed. This also could be interesting if a more systematic case comparison was made, but also that is not done. Also the methodological base is weak, with 1 intensively studied case, 2 medium studied cases, and 2 hardly studied cases. The explanation for this variation is not convincing. Herewith all possible interesting insights from these cases, are not analyzed nor presented. The discussion and conclusion also reflect this, because hardly any interesting patterns are found.
Finally, I have my doubts whether the paper fits the journal. This also related to the way the public value capture instruments and the use of NDO’s is presented. As sustainability reader, it is hard to understand these parts of the paper.
Author Response
Thank you for the paper on this interesting topic of developer obligations and their contribution to public values. On forehand, I was very much interested because a more public value driven view on this kind of instruments is really needed. Unfortunately, the paper was in my view not able to really fill this gap. Firstly because of the limited theoretical work on public value and secondly because of the thin case study which led to a descriptive instead of a comparative analysis. I will elaborate these points.
Author response: Thank you for urging me to develop the work with public value theory and the case analysis in the paper. It was encouraging to read that you find the topic interesting and worthy of further development. I have revised the paper following your comments below.
Firstly, the work on public value. Public value is an important concept in this study. It is not necessary to involve all work on public value. However, I miss in theoretical part some important scholars and discussions. You limit your understanding of public value creation to the work of Moore. His work is important but also discussed, especially they way he presents public values as objective measurable values and professionals as the actor to measure these values. All this kind of discussions including the work of other scholars (e.g. Bozeman, Meynhardy, Stoker) is missing.
Author response: Thank you for urging me to engage with more of the public value literature and improve this part of the paper. I am also very thankful for your suggestions for further reading. Section 2.3 on public value has been developed and now discusses work from more of the prominent scholars in the field, including those that you suggested. The discussions over public values as objective and measurable are now presented here, in addition to the literature on public value creation. Furthermore, the findings are now discussed (in section 6) in relation to more of this public value literature.
Secondly, the cases have the potential to be compared. However, the cases are only used to describe the way NDO’s could lead to public values. This could be interesting if also was analyzed whether this value was realized, but that is not analyzed. This could be interesting if more systematically was analyzed how the kind of NDO was related to the kind of public values, but that is not analyzed. This also could be interesting if a more systematic case comparison was made, but also that is not done.
Author response: Thank you making these suggestions to improve the analysis of the cases and develop the findings. I fully agree that the cases should be compared to each other more and have worked on developing this part of the analysis. In the results sections, more similarities and differences between the cases are now highlighted and discussed.
In response to your comment, I have also analysed the types of NDOs used for the different public value dimension further. As a result I was able to draw some conclusions in relation to achieving substantive, procedural and performance public values, which are presented in the discussion.
Since the district developments in this study are ongoing it is not yet possible to evaluate whether or not the municipalities’ public value creation objectives are actually realized. For this reason, the aim of the study is specifically limited to investigating the nature of the municipalities’ objectives in these types of district developments. However, I do agree that evaluating whether the public value they seek to create is actually realised is an important question and stress that future research should investigate this further in section 7.3.
Also the methodological base is weak, with 1 intensively studied case, 2 medium studied cases, and 2 hardly studied cases. The explanation for this variation is not convincing. Herewith all possible interesting insights from these cases, are not analyzed nor presented.
Author response: Thank you for noting this shortcoming. I have developed the justification for my selection of materials and in section 4.2. I explain more clearly why the amount of relevant content from each case is comparable.
The discussion and conclusion also reflect this, because hardly any interesting patterns are found.
Author response: Thank you for urging me to developed the discussion and conclusion and to look for more interesting patters to present. More patterns between the cases did emerge while working more on the analysis. These are now discussed in relation to the previous literature and theory in the discussion and conclusion.
Finally, I have my doubts whether the paper fits the journal. This also related to the way the public value capture instruments and the use of NDO’s is presented. As sustainability reader, it is hard to understand these parts of the paper.
Author response: Thank you for raising these concerns. The paper is primarily written for the special issue on “Public Value Capture” with that audience in mind, but should of course also be accessible for the wider Sustainability audience. I have gone through and worked on the wording to make these sections a bit easier to read. If there are any particular parts that are still unclear, I would more than happy to go back and rephrase them or provide more explanations.
Thank you for your constructive comments and valuable feedback.
Round 2
Reviewer 3 Report
Although the paper was revised on major points, I still have strong doubts about the paper. Firstly, the work on public values is added to the theoretical section, however it doesn’t had impact on the rest of the article. Also the analysis and conclusions are not improved by using this theoretical field. The second point, which is most important, is that there is still the problem of the cases. As mentioned before, the methodological base is weak and because of that the analysis does not lead to relevant outcomes. In this revised version explanations are added, but the real problem is not solved. The methodological base is still weak and the results are still descriptive without touching upon the interesting questions/topics which are presented in the introduction.
Author Response
Although the paper was revised on major points, I still have strong doubts about the paper. Firstly, the work on public values is added to the theoretical section, however it doesn’t had impact on the rest of the article. Also the analysis and conclusions are not improved by using this theoretical field.
Author response: Thank you for urging me to continue developing the work with public value theory in the paper. In response to the critique, this part of the theory specifically concerning public values is worked into more sections of the paper. For instance, the connection between the municipalities’ sustainability objectives and achieving public values is made more explicit and explored more (see e.g. section 3.1. line 290-291, section 3.3. line 352-354, section 5.1. lines 518-521, 579-580, section 5.2. line 624, paragraphs 3 and 4 in the discussion).
The second point, which is most important, is that there is still the problem of the cases. As mentioned before, the methodological base is weak and because of that the analysis does not lead to relevant outcomes. In this revised version explanations are added, but the real problem is not solved. The methodological base is still weak and the results are still descriptive without touching upon the interesting questions/topics which are presented in the introduction.
Author response: The reviewer’s feedback is appreciated and the method section has been revised. In response to the critique, the justifications for the methodological choices that were made have been developed with the hope that they are now communicated more clearly and effectively in the paper. Primarily, the justification for the multiple case study design is elaborated on in section 4.1. in connection to answering the two research questions. The strengths and weaknesses of this method are described in more detail. The case study method is well suited for, and often used for the purpose of, generating rich descriptions of empirical phenomenon, which is especially appropriate for answering the first research question in the paper. The aim of answering the first research question is to provide an explanation of how sustainability-oriented NDOs are used to create public value in sustainability-profiled districts, so these results should be descriptive in nature. The purpose of answering the second research question is to exemplify the types of public value outcomes municipalities pursue in these types cases, which the multiple case study method was considered suitable for as well. In addition to this, more of the measures taken to ensure the validity of results are presented and the cross-case analysis is also supported and described in more detail at the end of section 4.3. More of the significant differences between the cases, identified in the cross-case analysis, are also presented throughout the results.
As mentioned, the theoretical discussion (section 6) has been elevated to draw more on the previous work dealing with public values. In the discussion, the results are discussed in relation to the questions and topics presented in the introduction and theory sections. The contributions presented in the discussion and conclusions are however mainly geared towards the public value capture literature as this is the primary audience of the special issue.